# Multiply fully recyclable carbon fibre reinforced heat-resistant covalent thermosetting advanced composites

Yanchao Yuan[1], Yanxiao Sun[1], Shijing Yan[1], Jianqing Zhao[1], Shumei Liu[1], Mingqiu Zhang[2], Xiaoxing Zheng[1] & Lei Jia[1]

Nondestructive retrieval of expensive carbon fibres (CFs) from CF-reinforced thermosetting advanced composites widely applied in high-tech fields has remained inaccessible as the harsh conditions required to recycle high-performance resin matrices unavoidably damage the structure and properties of CFs. Degradable thermosetting resins with stable covalent structures offer a potential solution to this conflict. Here we design a new synthesis scheme and prepare a recyclable CF-reinforced poly(hexahydrotriazine) resin matrix advanced composite. The multiple recycling experiments and characterization data establish that this composite demonstrates performance comparable to those of its commercial counterparts, and more importantly, it realizes multiple intact recoveries of CFs and near-total recycling of the principal raw materials through gentle depolymerization in certain dilute acid solution. To our best knowledge, this study demonstrates for the first time a feasible and environment-friendly preparation-recycle-regeneration strategy for multiple CF-recycling from CF-reinforced advanced composites.

---

[1] College of Materials Science and Engineering, South China University of Technology, Guangzhou 510641, China. [2] Key Laboratory for Polymeric Composite and Functional Materials of Ministry of Education, School of Chemistry and Chemical Engineering, Sun Yat-sen University, Guangzhou 510275, China. Correspondence and requests for materials should be addressed to Y.Y. (email: msycyuan@scut.edu.cn) or to S.Y. (email: ysjshijingyan@163.com).

Continuous high-performance carbon fibre (CF)-reinforced high-performance thermosetting resin matrix advanced composites are characterized by specific strength, specific stiffness, size stability, designability and so on, and have become widely employed in many high-tech fields including aerospace, wind power, transportation and so on in recent decades[1,2]. Their widespread use, meanwhile, generates a substantial amount of waste of such CF-reinforced composites (CFRCs), including their off-cuts, unused prepregs and end-of-life components[3–6]. It is desireable to retrieve CFs nondestructively from these waste not only because of the high cost of CFs but also because of their consumption in great amounts in CFRCs (up to ~70 vol%) (refs 1,2). However, the inherent properties of thermosetting resin matrices, for example, irreversible three-dimensional cross-linked network structure, mechanical strength, thermal stability and chemical resistance and so on, do not allow them to be gently melted or dissolved as thermoplastic resins. Disposing CFRC scrap with traditional mechanical recycling methods, burning or burying would destroy CFs in a wasteful manner. Therefore, effective and nondestructive recovery of costly CFs and other useful raw chemicals from such thermosetting resin matrix advanced composites remains an important research goal.

To date, several recycling methods have been brought forward, including thermolysis and solvolysis[3–6]. However, even recently reported advanced recycling techniques would still destroy the size and braided order of CFs seriously as well as their surface structure in different degree[3–9], whereupon only two types of CFs, random short fluffy fibres and woven fabric pieces, can be retrieved[6]. Such reclaimed CFs can be used as substitute for virgin CFs in sheet molding compound and bulk molding compound materials, but they can never be reincorporated in the same high-tech applications where they are recovered from. This greatly depresses their commercial use and value, and the bottleneck problem is the contradiction between the requisite harsh degradation conditions of these traditional high-performance thermosets (for example, high temperature, high pressure, strong oxidation environment)[3–9] and their unavoidable harm on the important properties of CFs (for example, performance, length and architecture). Consequently, developing novel thermosetting matrix resins degradable under mild recycling conditions/processes offer a possible solution to this conflict. So far, three CFRC systems towards this goal have been built by creatively using relatively weak and unstable dynamic covalent structures (Table 1)[10–21]. The first one is the dynamic acetal [-O-CH(CH$_3$)-O-] epoxy system established by Hashimoto et al.[15], which can degrade easily via the acid hydrolysis of acetal linkages at room temperature and then generates undamaged but partially disordered CFs[15,16]. The second one is the dynamic polyimine [-C=N-] system brought up by Taynton et al.[17,18], which is able to degrade under the mild transamination reaction among excess diamine monomers and, therefore, realizes full recycle of woven CFs. Third one is the dynamic disulfide [-S-S-] epoxy system constructed by Luzuriaga et al.[19–21], which can degrade and recover CFs easily through thiol-disulfide exchange reaction. Compared with traditional epoxy thermosets (Table 1a), these three systems demonstrate obviously superior CF recyclability along with some other special attributes such as self-healing and malleability. However, due to the weak stability of dynamic covalent bonds[10–14], these novel composites exhibit moderate or lower performances on thermal stability, mechanical property and chemical resistance in comparison to traditional high-performance thermosets such as epoxy and bismaleimide (BMI) resins (Table 1b). This would limit their applications in high-tech fields, for example, microelectronics, transportation and aerospace industries, which are characterized by tough and varied working conditions and, therefore, are highly demanding

on material properties[1,2,22]. The abovementioned advances have shown that mild recovery process is crucial to preserve the CF structure, and thus inspire further development of CFRC recycling[15–21].

Besides these works, some other studies also concentrated on the design and synthesis of novel degradable thermosetting resins by using either dynamic or ordinary covalent bonds[23–35]. Although whether these new resins can produce high-performance CFRCs applicable in high-tech fields and meanwhile realize unimpaired withdrawal of CFs remains unclear, these contributions are important and guide the subsequent studies. Along these lines, García et al.[33] innovatively adopted hexahydrotriazine skeleton, which has strong covalent bonds but can be hydrolysed under certain mild acidic conditions, and successfully synthesized one poly(hexahydrotriazine) (PHT) thermosetting resin film using 4,4′-oxydianiline (ODA) and paraformaldehyde (PFA). This film, denoted as the ODA-PHT, demonstrates extremely high rigidity but unfortunately deficient toughness and heat resistance ($T_d \approx 238\,°C$), which necessitate further improvement. Results from this research suggest novel degradable thermosetting matrix resins with stable covalent structures as a particularly promising way to fundamentally solve the conflict between high composite performances and undamaged CF recovery. To the best of our knowledge, there is no published report having ideally settled this conflict up to now.

Based on this background, we designed a new synthesis scheme and successfully prepared one novel fully recyclable CF-reinforced PHT resin matrix advanced composite, which not only exhibits performance comparable to those of commercial epoxy and BMI advanced composites, but also allows multiple undamaged recoveries of CFs and nearly full recycling of the main raw materials via gentle depolymerization in special dilute acid solution (Fig. 1). As far as we know, this study for the first time provides an ideal, feasible and environment-friendly strategy for multiple intact CF recycling from high-performance CFRCs without compromising their properties.

## Results

**Development of one novel high-performance PHT resin.** In our study, the PHT resin was successfully prepared with 2,2-bis[4-(4-aminophenoxy)phenyl]propane (BAPP) and formaldehyde through a water-catalysed sequential addition condensation reaction and the followed curing processes (Fig. 2a,b), and therefore is denoted as the BAPP-PHT. The synthesis details of this resin can be referred to the methods section, and its complete characterization data are expatiated in Supplementary Note 1 and Supplementary Table 2. Briefly, its tensile strength, Young's modulus and plane-strain fracture toughness ($K_{Ic}$), respectively, reach 124.71 MPa, 4.76 GPa and 1.94 MPa·m$^{1/2}$ (Supplementary Figs 2–6), and indicate high strength, stiffness and toughness in combination with other attractive mechanical attributes (Table 1b); its high $T_g$ and $T_d$ (up to 200.1 and 368.5 °C, respectively, Fig. 2e,f) demonstrate attractive thermal stability, consistent with the strong stable C–N covalent bond and cross-linking structure of the hexahydrotriazine skeleton (Supplementary Fig. 7)[36]; In addition, the resistance experiments suggest that this resin can tolerate various representative solvents (for example, tetrahydrofuran (THF), acetone, dichloromethane, toluene, and so on.), oils, alkaline solutions, salt solutions, strong oxidizing agent solutions (for example, 30 wt% H$_2$O$_2$), and weak or some strong acid solutions (for example, 10 wt% H$_2$SO$_4$ and 37 wt% concentrated HCl) (Supplementary Table 3). On the whole, the BAPP-PHT demonstrates properties in line with those of common commercial high-performance epoxy and BMI resins (Table 1b)[1,2], and, therefore, can be regarded as a suitable resin matrix candidate for high-performance CFRCs.

**Table 1 | Performance parameters of the BAPP-PHT and the CF/PHT composites and a comparison with other analogous resins and composites. (a) Recyclability; (b) Thermal stability and mechanical property.**

(a)

| Samples | Functional covalent bond type | Recycling mechanism | Recycled CFs' braided structure | Recycled CFs' surface chemical structure | Recycled CFs' mechanical property | Recovery of matrix main raw materials |
|---|---|---|---|---|---|---|
| The traditional epoxy/CF[7] | Ordinary covalent bonds | Oxidative degradation under mild conditions with $H_2O_2$ | Completely destroyed | Slightly damaged | Slightly declined | No |
| The traditional epoxy/ 67.5 vol% CF[8] | Ordinary C–N | Selective cleavage of C–N bonds with $AlCl_3/CH_3COOH$ at 180 °C for 6 h | Completely destroyed | Slightly damaged | Slightly declined | No |
| The acetal epoxy/62 vol% cross-ply CF[15,16] | Dynamic acetal | Easy acid-induced hydrolysis of dynamic acetal linkages | Partially disordered | Unchanged | Unchanged | No |
| The polyimine /60– 75 wt% twill weave CF[17,18] | Dynamic C=N | Dynamic C=N bonds exchange among excess diamines | Basically reserved | Not reported | Not reported | Yes |
| The disulfide epoxy/ plain weave CF[19,20] | Dynamic S–S | Dynamic S–S bonds exchange in mercaptans | Basically reserved | Not reported | Not reported | No |
| The cross-ply CF/PHT composites* | Ordinary C–N | Hydrolysis of ordinary C–N bonds in special acid mixed solution | Almost unchanged | Unchanged | Unchanged | Yes |

(b)

| Samples | $T_g$ (°C) | $T_d$ (°C) | Tensile strength (MPa) | Young's modulus (GPa) | Elongation at break (%) | Flexural strength (MPa) | Flexural modulus (GPa) | Compression strength (MPa) | Short-beam strength (MPa) | Fracture toughness (MPa·m$^{1/2}$) |
|---|---|---|---|---|---|---|---|---|---|---|
| The BAPP-PHT | 200.1 | 368.5 | 124.7 | 4.8 | 3.8 | 151.6 | 4.0 | 162.7 | — | 1.94 |
| TGDDM/DDS[†] | ~220 | ~306 | 58 | 4.2 | 1.6 | 125 | 4.0 | 178 | — | 0.71 |
| HexFlow RTM6[‡] | ~215 | ~300 | 75 | 2.9 | 3.4 | 132 | 3.3 | — | — | 0.64 |
| XU292[§] | ~310 | ~400 | 93.3 | 3.9 | 3.0 | 184 | 3.98 | 209 | — | 0.70 |
| The unidirectional CF/PHT composites[||] | 199.5 | 384.2 | 1806.6 | 141.7 | 1.4 | 1241.2 | 127.4 | 343.3 | 69.1 | — |
| The cross-ply CF/ PHT composites[||] | 198.2 | 376.3 | 741.2 | 68.3 | 1.2 | 829.7 | 54.8 | 351.5 | 75.5 | — |
| T300/914[¶] | ~190 | ~310 | 631 | 70 | — | 912 | 61 | — | 64 | — |
| T300/5405[#] | ~210 | ~380 | 1,787 | 147.8 | — | 1,560 | 108 | — | 96.8 | — |
| The acetal epoxy[15,16] | 21–110 | 225–273 | 52.5–65.4 | 1.2–3.0 | 3.7–8.5 | — | — | — | — | — |
| The acetal epoxy/62 vol% cross-ply CF[15,16] | — | — | 1,343–1,375 | 66.8–73.9 | 1.75–1.82 | — | — | — | — | — |
| The polyimine[17,18] | 18–135 | — | 10–64 | 0.13–1.0 | 5–150 | — | — | — | — | — |
| The polyimine/60– 75 wt% twill weave CF[17,18] | — | 268–305 | 148–399 | 12.2–15.5 | 1.0–3.8 | 142–255 | 32.4–38.1 | — | — | — |
| The disulfide epoxy[19,20] | ~130 | ~300 | 88 | 2.6 | 7.1 | — | — | — | — | — |
| The disulfide epoxy/ plain weave CF[19,20] | Not reported | — | — | — | — | — | — | — | — | — |

*The plain cloth stacking sequence used was cross-ply [0/90]$_n$ and the CF content in the composites was 50.1 vol%. Mechanical tests were carried out along the direction of 0° or 90° and the results showed no difference.
†TGDDM/DDS stands for the tetradiglycidyl diaminodiphenylmethane/diaminodiophenylsulfone epoxy resin system.
‡HexFlow RTM6 is one high-performance mono-component resin of Hexcel Corporation and is made up of TGDDM, methylene-bis-(2,6-diethylaniline) and methylene-bis-(2,6-diisopropylaniline).
§XU292 is one high-performance BMI resin of Ciba-Geigy, which is one bis(4,4'-maleimidodiphenyl)-methane toughened by 2,2-bis(4-hydroxy-3-allylphenyl) propane.
||Unidirectional cloth stacking sequence used was [0]$_n$ and the CF content in the composites was 61.2 vol%. The test results are from the 0° direction samples.
¶T300/914 is one high-performance advanced composite of Hexcel Corporation, which is composed of 5 satin T300 CF cloth and HexPly 914 epoxy matrix.
#T300/5405 is one high-performance advanced composite of AVIC Beijing Institute of Aeronautical Materials, which comprises unidirectional T300 CF cloth and BMI 5405 matrix. The properties of BMI 5405 are similar to that of Narmco 5245C BMI resin.

**Preparation and characterization of the CF/PHT composite.** In our study, the preparation of CF/PHT composite is divided into two steps: prepreg preparation and the subsequent compression molding (Fig. 1). The first step was fulfilled by prepolymer solution impregnation method, and the resin content in prepregs was adjusted by the prepolymer concentration and the size of CF cloth. Two kinds of prepregs were prepared individually using plain weave CF cloth and unidirectional one, and their monolayer thicknesses, CF contents (calculated with ASTM D3529M-10 standard), and volatiles contents (calculated with ASTM D3530-97(2015) standard) are 0.125 and 0.221 mm, 57.1 wt% and 60.2 wt%, and 1.8 wt% and 1.5 wt%, respectively. The second step was accomplished with hand lay-up and compression molding techniques. In order to clearly trace the whole recycling process, one virgin plain CF cloth with artificially irregular edges and a white lock glass line (Fig. 1) was used as the outermost layer of the composite laminate, thickness of which can be easily adjusted by the prepreg layer number (Fig. 2c,d). The density and fibre content of the obtained cross-ply CF/PHT composite laminate and unidirectional one are 1.504 and 1.586 g cm$^{-3}$ (ASTM D792-13 standard), and 50.1 vol% and 61.2 vol% (ASTM D3171-15 standard), respectively.

After preparation, the fracture appearance of these composite laminates was investigated and no visible voids or defects were found on their cross sections and fracture planes (Supplementary Fig. 9); And their tensile and flexural tests show that the matrix resin entirely filled in and tightly combined with the CF bundles (Supplementary Fig. 9c–f), which is beneficial to their interior load transfer. These results preliminarily suggest a successful preparation process. Subsequently, the mechanical properties of these two CF/PHT composites were characterized according to corresponding ASTM standards and the results are shown in Table 1b and Supplementary Figs 10–12. Compared with T300/ 914 (5 satin T300 CF cloth and HexPly 914 epoxy matrix), a representative commercial high-performance epoxy resin advanced composite of Hexcel Corporation, our cross-ply CF/

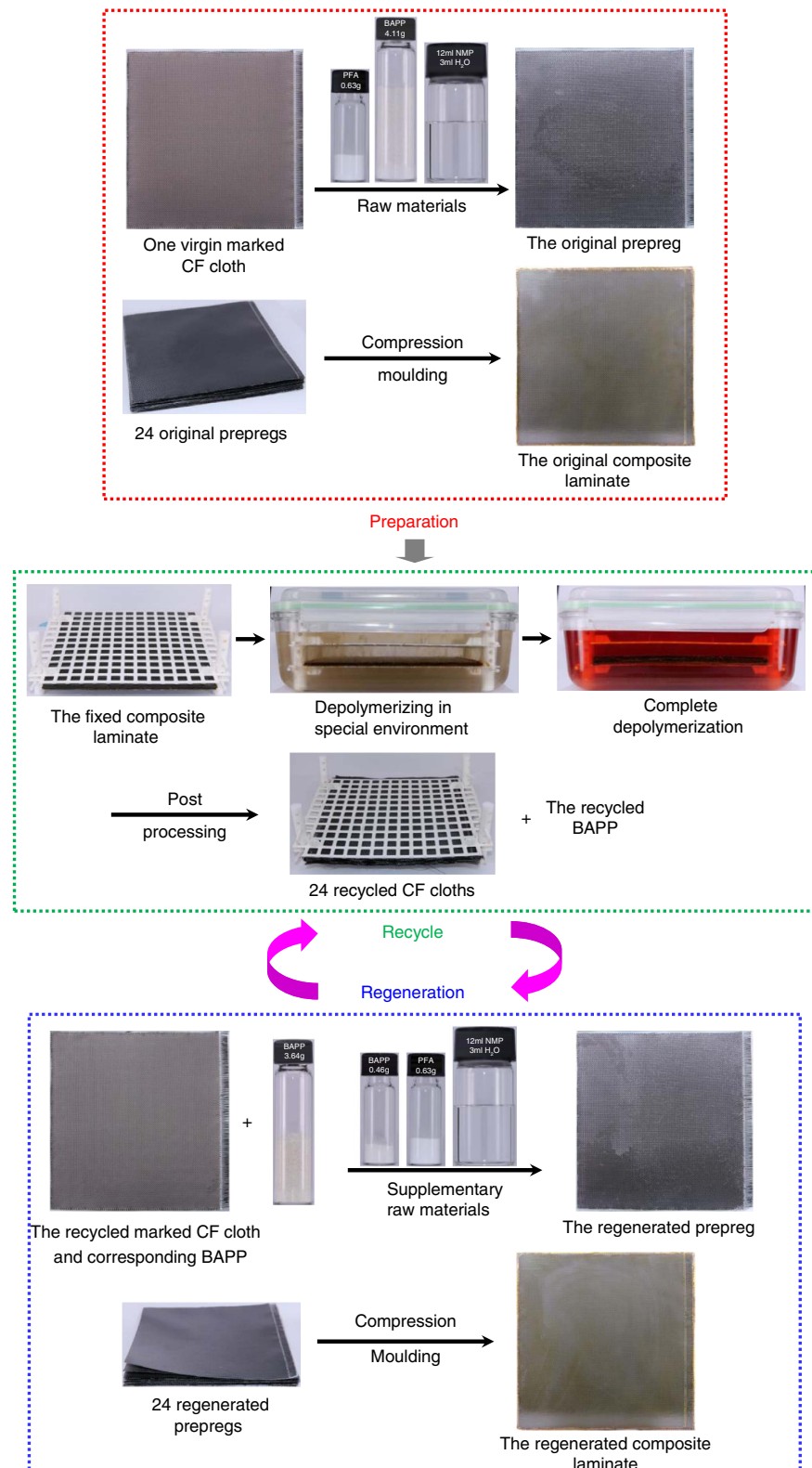

**Figure 1 | The recycle and regeneration processes of the CF/PHT composite.** A total of 24 virgin plain weave CF cloths (20 × 20 cm) were used to prepare the prepreps, and one virgin CF cloth with irregular edges and a white lock glass line is used as the outermost layer of the composite laminate to clearly trace the whole process.

PHT composite demonstrates roughly the same mechanical performance parameters but much higher heat resistance ($T_d = 376.3$ °C, Fig. 2e and Table 1b). Similarly, our unidirectional CF/PHT composite also displays almost identical mechanical characteristics to an analogous marketed product, T300/5405 (unidirectional T300 CF cloth and 5405 BMI resin)

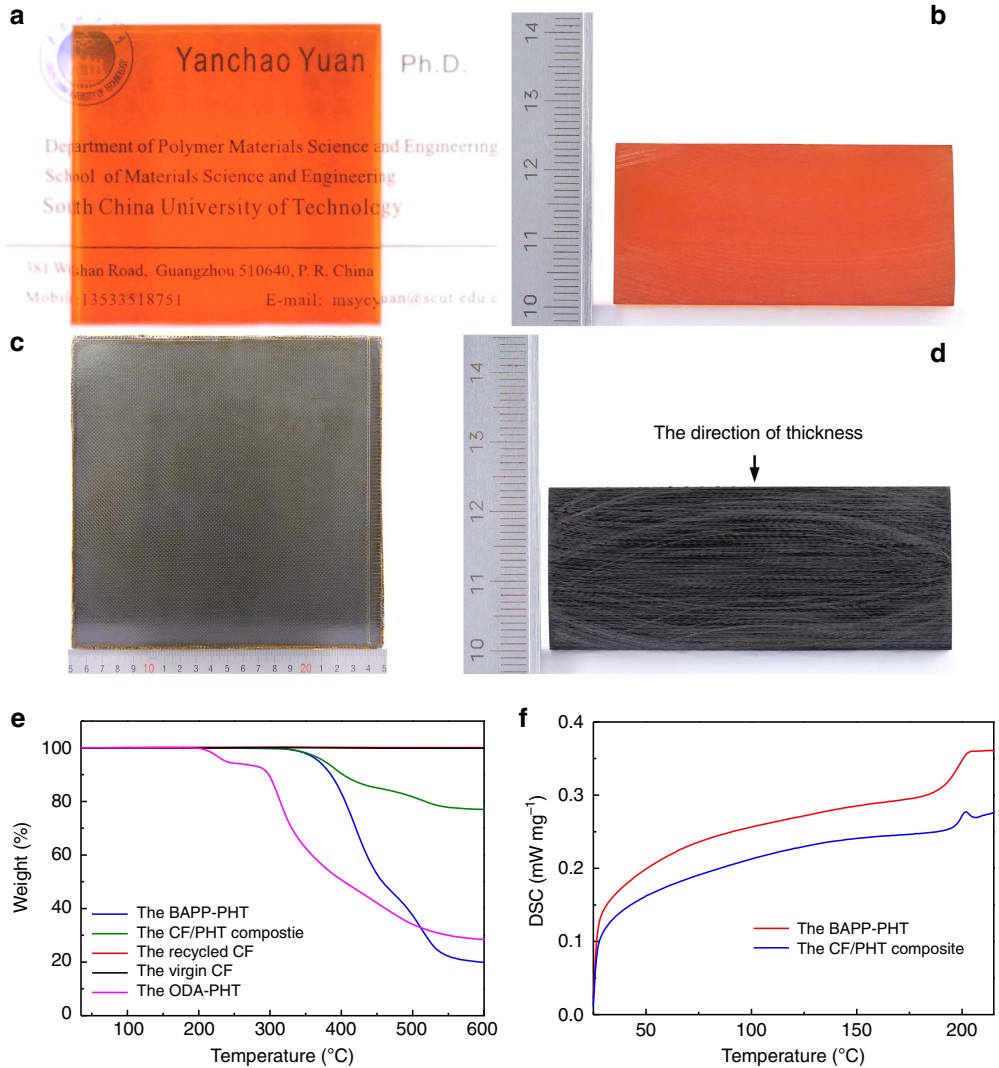

**Figure 2 | Apperance and performance of the BAPP-PHT and the CF/PHT composites.** (**a**) The BAPP-PHT specimen with a nominal size of 5 cm × 5 cm × 24 mm (L × W × H) shows good transparency on a namecard; (**b**) the polished cross section of the BAPP-PHT; (**c**) One representative cross-ply laminate at the nominal size of 20 cm × 20 cm × 3 mm (L × W × H); (**d**) One representative cross-ply laminate with the nominal size of 4 cm × 6 cm × 23.7 mm (L × W × H) and the cross section been polished with sandpapers; (**e**) TGA curves of the BAPP-PHT, the cross-ply CF/PHT composite, the recycled CF, the virgin CF and the ODA-PHT. The ODA-PHT was prepared with the same synthesis conditions as the BAPP-PHT with the dosages of ODA and PFA adjusted accordingly. (**f**) DSC results of the BAPP-PHT and the cross-ply CF/PHT composite. The digital unit of the rulers is 1 cm.

high-performance BMI resin advanced composite (Table 1b). In addition, measured with the compression test method described for thin composite laminates in ref. 37, our CF/PHT composite laminates present much the same compressive strengths (343.3 and 351.5 MPa, respectively, Table 1b and Supplementary Fig. 12) as woven AGP193-PW/8552 (10 plies) and cross-ply AS4/3051-6 (lay-up $[0/90]_{3S}$), two well-known advanced CF/epoxy composites of Hexcel Corporation used for aeronautical applications[37]. Overall, these results preliminarily indicate that the mechanical and thermal properties of our CF/PHT composites are comparable or even superior to those of currently prevailing commercial high-performance epoxy and BMI resins matrix advanced composites. Chemical resistance endowed by the PHT matrix resin further strengthen the overall relative performance among related systems.

**Degradation and recovery of the BAPP-PHT.** The hexahydrotriazine ring of PHT resin can be fractured or hydrolysed under

the action of H$^+$ (ref. 33). Hence in theory, the BAPP-PHT should also be able to degrade under acidic conditions, just as the ODA-PHT[33]. However, as abovementioned, the BAPP-PHT is recalcitrant to strong acids (Supplementary Table 3), implying that its contact with H$^+$ and the subsequent acidolysis might be hindered by some factors. Contact angle ($\theta$) is an important wettability indicator of solid by certain liquid, and when $\theta < 90°$, the solid can be wetted by the liquid and the smaller is $\theta$, the better is the wettability, while when $\theta > 90°$, the liquid can neither stay on the solid surface nor contact its inside. In our study, static $\theta$ of certain solvent on PHT resin was determined by sessile-drop method at room temperature, and herein we replaced strong acid solution with distilled water to protect the test instrument away from acid corrosion. The $\theta$s of water on the ODA-PHT and the BAPP-PHT are 86.9° and 104.7°, respectively, which signifies that water solution can wet the former but not the latter and therefore reasonably explains the latter's high tolerance to strong acid solution. However, unlike water, organic solvents such as THF, acetone, etc. display small $\theta$s (for example, 6.7° for THF) and

good wettability on the BAPP-PHT. In view of this, the wettability of water and THF mixture on the BAPP-PHT was further tested. As shown in Fig. 3b, $\theta$ value of the mixture linearly decreases with its THF proportion, indicating that THF can greatly improve the wettability of water solution on this resin. Based on this, we prepared a series of 1 M HCl solutions with different proportions of THF and used them to tentatively decompose the BAPP-PHT. According to Fig. 3a, the resin can be degraded in certain solutions and the breakdown process apparently demonstrates two stages, dissolution and precipitation, which are correspondingly speculated as its depolymerization into soluble oligomers and further generation of raw material

monomers. Here we define the time from acid addition to the resin's complete dissolution as its first-stage depolymerization time ($t_1$). Seen from Fig. 3b, $t_1$ in certain 1 M HCl/THF solution is closely correlated to the wettability of its corresponding $H_2O$/THF solution on the BAPP-PHT. To be specific, when the $\theta$ of $H_2O$/THF solution varies from 56.9° ($H_2O$/THF = 7/3) to 9.0° ($H_2O$/THF = 2/8), $t_1$ in the equivalent 1 M HCl/THF solution decreases sharply from 35.2 to 3.1 h; As for $H_2O$/THF solutions with higher $\theta$s, for example, 68.5° ($H_2O$/THF = 8/2) and 85.4° ($H_2O$/THF = 9/1), their corresponding 1 M HCl/THF solutions can only partially dissolve the resin (7-day mass loss: 70.2% and 23.8%, respectively); And the resin showed no

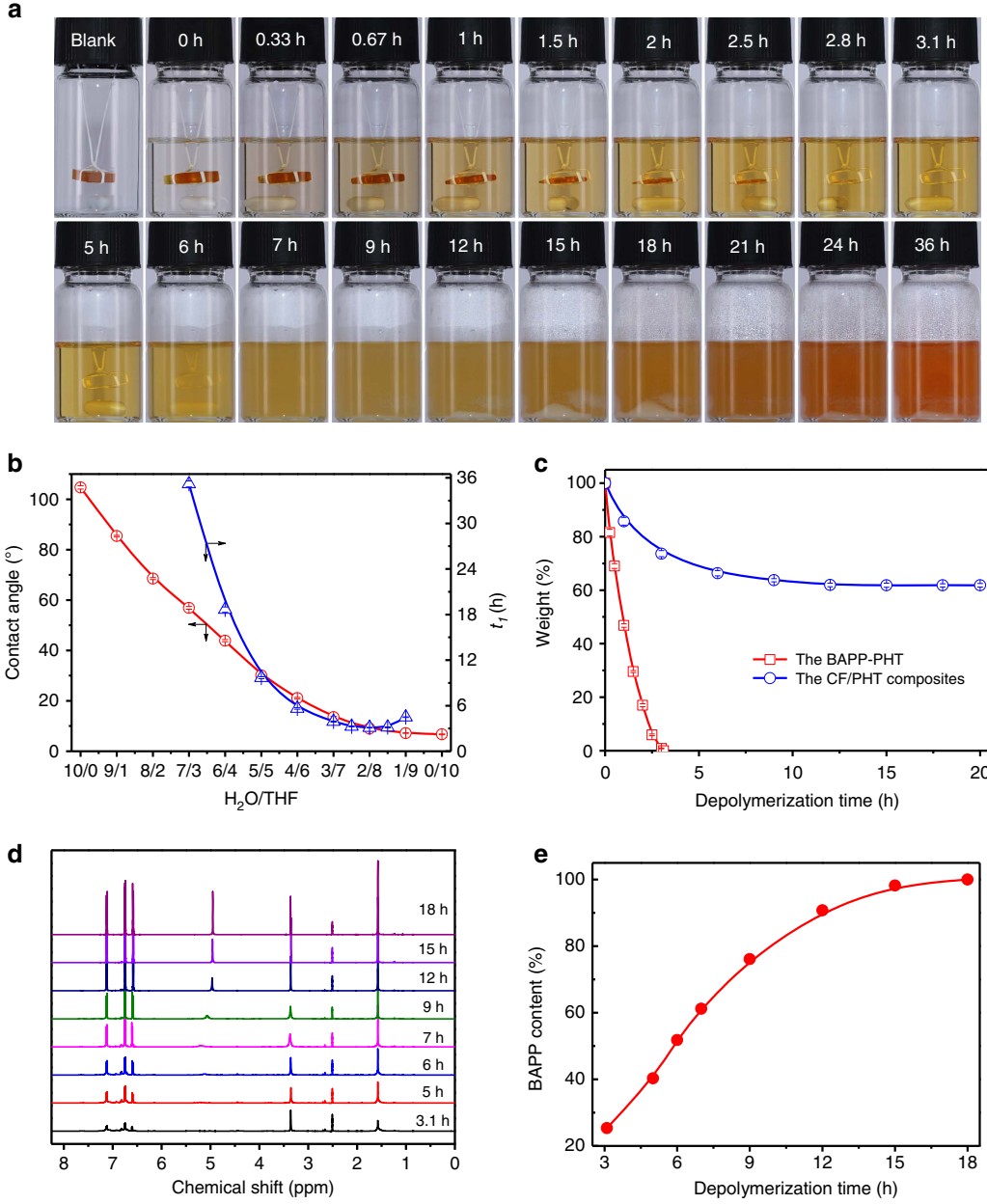

**Figure 3 | Depolymerization and recovery of the BAPP-PHT at room temperature.** (**a**) The resin's degradation status in 10 ml 1 M HCl/THF solution at different times; (**b**) The resin's first-stage depolymerization time ($t_1$) in different 1 M HCl/THF solutions and the contact angles of their corresponding $H_2O$/THF solutions; (**c**) The depolymerization kinetics of the BAPP-PHT and the CF/PHT composite in 1 M HCl/THF solution (specimen size: 1 cm × 1 cm × 3 mm); (**d**) $^1H$ NMR spectra of the depolymerized products of the BAPP-PHT in 1 M HCl/THF solution at different times; (**e**) The generation curve of BAPP.

observable changes after 7 days in 1 M HCl solution, in accordance with the higher $\theta$ than 90° (104.7°) of water. Replacing THF with acetone and/or HCl with $H_2SO_4$ produced the same dependence result, indicating the crucial role of solution wettability in the BAPP-PHT depolymerization. But interestingly, after $t_1$ reaches the minimum at 1 M HCl/THF ($H_2O$/THF) = 2/8 solution (3.1 h, Fig. 3b), further increasing the THF proportion (for example, 8.5 or 9 tenths) contrarily extends the time even though the solution wettability is improved meanwhile. Considering the complete BAPP-PHT depolymerization in this system only theoretically needs ∼0.03 ml water, suggesting that water has a role beyond simply driving hydrolysis in this process, for example, proton transmitter or hydrolysis promoter. Therefore, it is necessary to maintain certain amount of water to gain the optimum depolymerization result. Then we fixed the THF proportion at 8 tenths and investigated the influence of HCl concentration ranging between 0.01 and 2.4 M (The maximum concentration can be prepared with 12 M concentrated HCl solution at this THF ratio, Supplementary Fig. 13) on $t_1$. When the HCl concentration varies from 0.01 to 0.1 M, the BAPP-PHT cannot be completely decomposed in 7 days, and its 7-day mass loss increases from 0 (0.01 M HCl) to 16.3% (0.03 M HCl) and then to 63.6% (0.1 M HCl). This illustrates the low efficiency of dilute acid solutions in depolymerizing the BAPP-PHT. However, the BAPP-PHT can be completely dissolved in 37.3 h as the HCl concentration increases to 0.2 M, and hereafter, its $t_1$ keeps decreasing with the continual increase of the HCl concentration, bottoms at 1 M HCl (3.1 h), and then slowly rises again. Besides, solutions with higher HCl concentrations (2.4 to 12 M) were also prepared using 12 M concentrated HCl solution and THF and experimented likewise for comparison (Supplementary Fig. 14); The results suggest that the $t_1$ value gradually extends from 4.9 to 70.2 h as the HCl concentration increases from 2.4 to 8.4 M, and then to >7 days in 9.6, 10.8 and 12 M HCl solutions (the corresponding 7-day mass losses of the resin are 50.7 wt%, 12.5 wt% and 1.2 wt%, respectively). Overall, 1 M HCl/THF ($H_2O$/THF) = 2/8 is the most efficient solution in the first-stage depolymerization of the BAPP-PHT.

As shown in Fig. 3a, after the BAPP-PHT was entirely dissolved, the clear solution became turbid and a grey white precipitate appeared. The precipitate was filtered, washed with THF and dried immediately for chemical structure identification. For comparison, BAPP hydrochloride was simultaneously prepared likewise, that is, dissolving pure BAPP in THF, then adding excess 1 M HCl solution to precipitate and further filtering and drying[38]. The NMR results show that, compared to BAPP, both the amino group ($NH_3^+$) and aromatic ring (Ar-H) of BAPP hydrochloride exhibit proton chemical shifts to lower field and the former is dramatically shifted from 4.9 ($NH_2$ of BAPP) to 10.3 p.p.m. (Supplementary Fig. 15). Similarly, the fourier transform infrared spectrum of BAPP hydrochloride (Supplementary Fig. 16) displays that the stretching vibration peaks of its $NH_3^+$ group is remarkably shifted from ∼3,300 cm$^{-1}$ ($NH_2$ of free amine) to ∼2,570 cm$^{-1}$, and meanwhile the in-plane bending vibration peak intensity of this group is obviously weakened at 1,611 cm$^{-1}$ in comparison with BAPP. Obviously, the grey white precipitate demonstrates almost the same NMR and fourier transform infrared spectra as BAPP hydrochloride and consistent changing rules with other aromatic amines[38,39]. Therefore, it is reasonably identified as BAPP hydrochloride. In the same way, the recovered product from the precipitate further treated with saturated sodium bicarbonate solution and water is recognized as BAPP (Supplementary Figs 15 and 16). Besides, the existence of formaldehyde in abovementioned filtrate was also verified with acetylacetone spectrophotometric method (HJ 601-2011 standard). That is to say, the BAPP-PHT is eventually depolymerized into its raw materials, BAPP and formaldehyde. In order to further clarify the generation process of BAPP, depolymerization solutions (1 M HCl/THF ($H_2O$/THF) = 2/8) of the resin at different times were neutralized with saturated sodium bicarbonate solution, filtered, rinsed and dried sequentially, then the obtained products were performed [1]H NMR analysis using the methyl proton peak area near 1.55 p.p.m. as internal standard (Fig. 3d) and the data were compared with pure BAPP to calculate its relative content. The outcomes show that, after the BAPP-PHT is completely dissolved at 3.1 h, the abundance of BAPP gradually increases with the depolymerization time, peaks at 18 h and stabilizes thereafter (Fig. 3e). The whole generation stage of BAPP takes about 14.9 h and its recovery reaches 94.7%. These findings illustrate the strong recyclability of the BAPP-PHT and its related composites.

**Recycling of the CF/PHT composite.** As mentioned above, the CF recyclability of CF/PHT composites is governed by the degradability of PHT resins. Here we demonstrated the recovery process of our cross-ply CF/PHT composite laminates in two frequently-used sizes, 1 cm × 1 cm × 3 mm and 20 cm × 20 cm × 3 mm ($L \times W \times H$), which are respectively used to study the recycling mechanism and testify the recovery effect of practical application (Figs 4 and 5). As illustrated in Fig. 4a, after 10 ml 1 M HCl/THF ($H_2O$/THF) = 2/8 solution was added, the small laminate was observed expanding and loosing continually in thickness direction until the matrix was completely dissolved. During this process, the CF cloths were taken out at various times and prepared for morphological observation, and the corresponding depolymerization solution were sequentially neutralized, filtered, cleaned and dried to gather the depolymerization product. The scanning electron microscopy (SEM) images in Fig. 4b clearly indicate that the BAPP-PHT is gradually depolymerized from the outside of the CF bundles to their inside. Seen from the degradation kinetic curve of the composite (Fig. 3c) and the X-ray photoemission spectroscopy (XPS) spectra of CFs (Supplementary Fig. 17), the complete depolymerization time of the BAPP-PHT is ∼20 h. Taking into account the generation time of BAPP, the total degradation process is extended to 36 h, and the BAPP recovery reaches 90.2% at this time.

Similarly, after submerged by 3 l 1 M HCl/THF ($H_2O$/THF) = 2/8 solution, the CF cloths in the big laminate also exhibited gradually vertical expansion and layer internal increase during the 36 h degradation process (Supplementary Figs 18 and 19). In the same way, the CF cloths and depolymerization products of the big laminate were also regularly separated and dried for subsequent measurements. Seen from Fig. 5, the retrieved CF cloths show hardly any apparent change in architecture and length except for a little edge perturbation. Their structure and performance were further characterized by SEM, XPS, XRD and monofilament tensile test method, and no perceptible changes were detected in their woven structure, surface morphology, chemical structure and mechanical properties (Fig. 5 and Supplementary Figs 20–23). Meanwhile, 88.6% of BAPP was reclaimed during this process. After adding a small amount of BAPP and PFA, the recovered BAPP and CF cloths were used to prepare the first batch of regenerated composite laminates (Fig. 1). These rebuilt laminates were processed into short-beam strength and flexural specimens according to ASTM D2344/D2344M-13 and ASTM D790 standards, and one small piece of them was used to recycle CF cloths and analyse for fibre structure. As shown in Fig. 5c, the once-regenerated composite laminate shows nearly the same short-beam strength, flexural strength and modulus as the original one. This should be attributed to the mild CF recycling process without significant

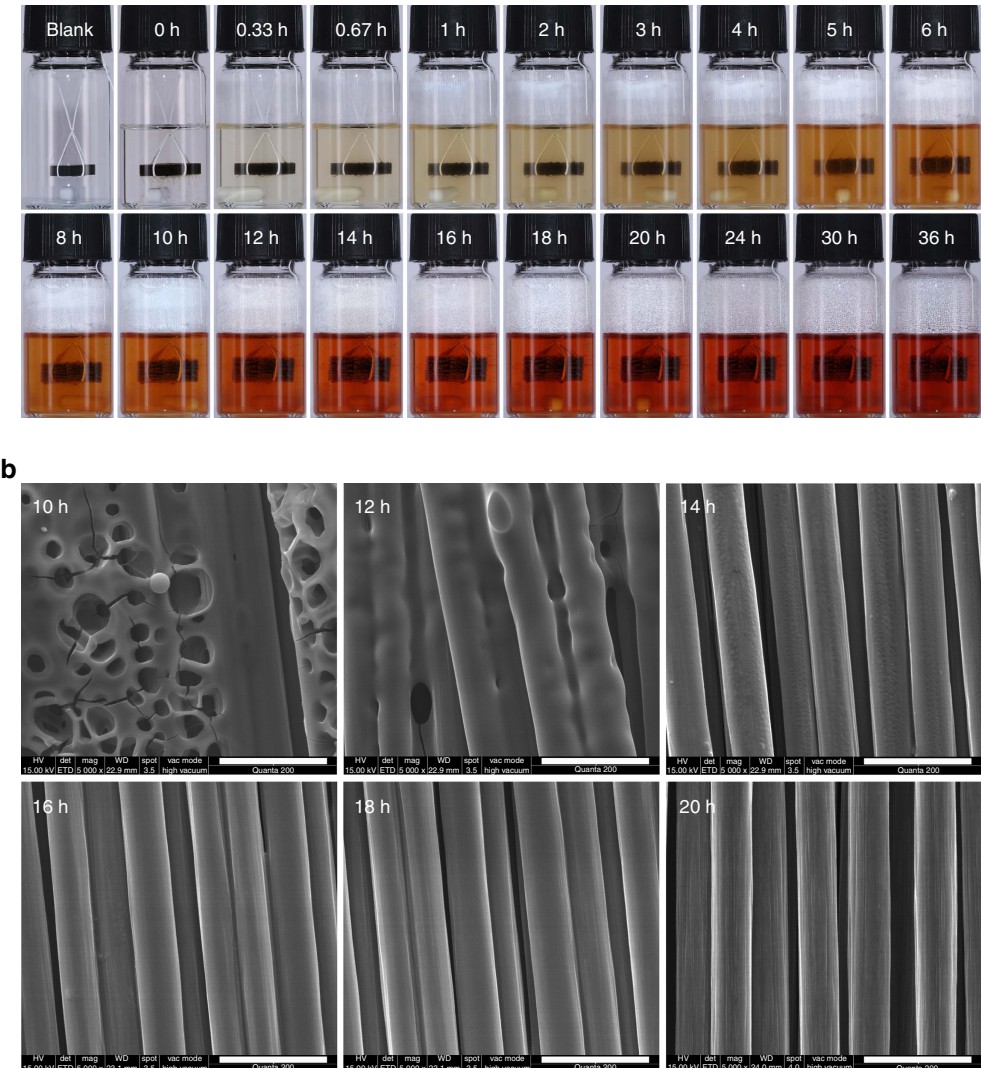

**Figure 4 | Degradation of the cross-ply CF/PHT composite laminates at room temperature.** (**a**) The degradation process in 10 ml 1 M HCl/THF solution at different times; (**b**) SEM images of the recycled CFs at different depolymerization times. Scale bar, 20 μm.

detriment to its architecture, length and performance. In fact, THF solutions with higher HCl concentrations can also completely degrade the laminate at room temperature without weakening any properties of the recycled CFs and the regenerated composite, but merely need longer degradation time (for example, 48 h for 3.6 M HCl/THF ($H_2O$/THF) = 3/7 and 60 h for 4.8 M HCl/THF ($H_2O$/THF) = 4/6) and impose certain corrosion on the white lock glass fibre of the outmost CF cloth (Supplementary Figs 24 and 25).

On this basis, the second and third recycling of the CF/PHT composites were sequentially performed, and the structures and properties of the twice- and thrice-recycled CFs and the regenerated composites were characterized, respectively. Seen from Fig. 5 and Supplementary Figs 20–23, just as the first one, the second and third recycling circulations barely affect the architecture, length and performance of the CF cloths and the composite mechanical properties. Moreover, the XPS results reflect that no chemical interaction occurs between the PHT matrix resin and the CF cloths (Supplementary Figs 21 and 22). Based on these results, we believe this recovery-regeneration scheme to be a feasible method for the multiple recycling of CF/PHT composites.

## Discussion

In summary, our experimental results and characterization data suggest that the CF/PHT thermosetting resin matrix advanced composite we developed possesses not only high performance but also efficient and nondestructive recoverability for multiple cycles of CFs and raw materials. Our findings verify that, on the one hand, it is feasible to use degradable resin with stable and strong covalent structure to develop CFRCs simultaneously with high performance and good recyclablity, and on the other hand, mild degradation conditions facilitate realizing multiple undamaged recycling of CFs from CFRCs without compromising their properties, and the key issue for this process is the wettability (that is, contact angle) of their depolymerization solution. Overall, this work demonstrates a practically relevant preparation-recycle-regeneration solution to the conflict between the high composite performance and the nondestructive CF recovery.

## Methods

**Materials.** Plain weave CF cloth (product designation: CO6142) and unidirectional CF cloth (product designation: UT-20G) were purchased from Toray Industries, Inc. (Tokyo, Japan), and the fibre details are listed in Supplementary Table 1. Before use, the cloths were soaked in acetone for 48 h to clean the surface,

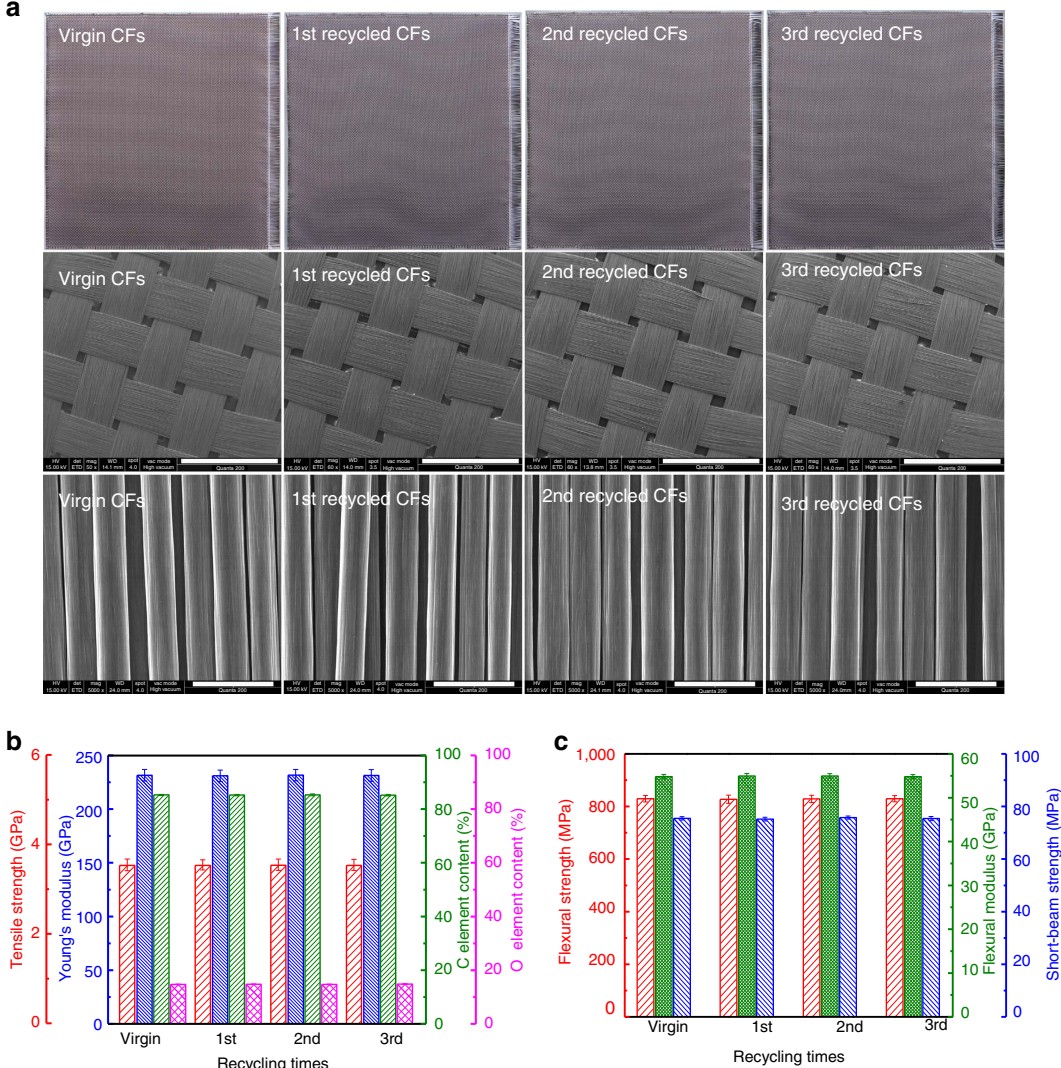

**Figure 5 | Multiple recycling and regeneration of the cross-ply CF/PHT composite laminates.** (**a**) The photos and SEM images of virgin and recycled CFs (the scale bars for second and third row SEM images are 2 mm and 20 μm, respectively); (**b**) The monofilament tensile properties and surface element contents of virgin and recycled CFs; (**c**) The flexural properties and short-beam strengths of original and regenerated cross-ply CF/PHT composite laminates.

and then dried at 80 °C for 6 h. Reagent-grade PFA was purchased from Sigma-Aldrich (St Louis, USA). BAPP and 4,4′-Oxydianiline (ODA) were purchased from Alfa Aesar (Ward Hill, USA), then purified and dried in a vacuum drier overnight before use. N-methyl-2-pyrrolidone (NMP) and THF were purchased from Guangzhou Chemical Reagent Factory (Guangzhou, China) and purified by vacuum distillation. Other high purity reagents and solvents were used without further purification.

**Preparation of the PHT resin with BAPP.** The polymerization mechanism is elucidated in Supplementary Figs 26–32 and Supplementary Note 3. The preparation process of the BAPP-PHT is divided into two steps: preparation of partially cured B-stage resin sheets and further compression molding. Typically, 63 mmol PFA, 27 ml NMP and 3 ml distilled water were added into a flask and continuously magnetically stirred at 80 °C for 0.5 h. Then the transparent formaldehyde solution was cooled down to 50 °C and 30 mmol BAPP was introduced. After 1 h reaction, the prepolymer solution was spread onto a pre-cleaned glass plate with polytetrafluoroethylene (PTFE) frame, then precured and removed solvents sequentially at 50 °C for 6 h, 80 °C for 2 h and 120 °C for 2 h. After that, the transparent B-stage resin sheet with a thickness of ∼0.3 mm peeled off from the glass surface. To meet the final product size, certain amount of B-stage resin sheets were put into enclosed molds with Frekote 700-NC mold-release agent and performed hot-pressing process. They were pressured to 3 MPa and continuously heated at 5 °C min$^{-1}$ to 200 °C for 2 h. After cooled down, the BAPP-PHTs with different sizes were obtained.

**Preparation of the CF/PHT prepregs.** As for plain weave CF cloth, 16.8 mmol PFA, 13.5 ml NMP and 1.5 ml distilled water were added into a flask and continuously magnetically stirred at 80 °C for 0.5 h. Then the transparent formaldehyde solution was cooled down to 50 °C and 8 mmol BAPP was introduced. A pre-cleaned glass plate with PTFE frame covered with a piece of CF cloth was prepared in advance. After prepolymerizing for 1 h, the solution was spread onto the CF cloth, then degassed for 2 min. The prepolymer was further precured and removed solvents sequentially at 50 °C for 6 h, 80 °C for 2 h and 120 °C for 2 h. After that, the CF/PHT prepreg peeled off from the glass surface. The preparation process for unidirectional CF cloth is the same but the amounts of raw materials were adjusted accordingly.

**Preparation of the CF/PHT composite laminates.** The CF/PHT composite laminates were manufactured by hand lay-up and compression molding techniques. CF/PHT prepregs were put into closed molds with Frekote 700-NC mold-release agent for hot-pressing process. They were pressed to about 3 MPa and continuously heated to 200 °C for 2 h. After cooling down, the composite laminates with different sizes were obtained. Using plain weave CF cloths, the cross-ply laminates with different thicknesses were prepared by the lay-ups: cross-ply [0/90]$_n$. Using unidirectional CF cloths, only unidirectional laminates with different thicknesses were prepared along 0° direction. After cutting the laminate edges and angle with a diamond saw, all composite test specimens were obtained by cutting along 0° or 90° direction, grinding and polishing.

**Recycling of the BAPP-PHT.** Similar to its resistance evaluation process to chemical reagents, the BAPP-PHT specimen with the nominal size of 1 cm × 1 cm × 3 mm (L × W × H) was bundled with PTFE wire and suspended in the middle of a glass bottle with a small magnet (Supplementary Fig. 8 and Supplementary Note 2). Acid solution was added into the bottle to submerge the specimen, and then was stirred slowly to depolymerize the resin gradually. The depolymerization solution was neutralized with saturated sodium bicarbonate solution, then the precipitate was obtained by filtering, cleaning and drying.

**Recycling of the CF/PHT composites.** The recovery process for the cross-ply CF/PHT composite was demonstrated with two kinds of specimens in different sizes, 1 cm × 1 cm × 3 mm and 20 cm × 20 cm × 3 mm (L × W × H), respectively. Among them, the small laminate was loosely bundled with PTFE wire and suspended in the middle of a glass bottle with a small magnet. Certain dilute acid solution was added into the bottle to submerge the specimen, and then stirred slowly to gradually depolymerize the composite matrix resin. During depolymerization, CF cloths with partially or completely depolymerized matrix were regularly taken out from the bottle, cleaned sequentially with saturated sodium bicarbonate solution, THF and distilled water, dried at 80 °C for 3 h, and then performed structure characterization. At the same time, the corresponding depolymerization solution was combined with the CF-cloth cleaning liquid and neutralized with saturated sodium bicarbonate solution, and then filtered, cleaned and dried to gather the precipitate. The big laminate was loosely fixed with a PTFE fixture with freely adjustable grid space, then was placed in a sealable glass vessel with appropriate size. Certain dilute acid solution was added into the vessel to submerge the specimen and the whole depolymerization process was illustrated in Supplementary Fig. 19. After the matrix resin was completely depolymerized, the CF cloths were taken out of the vessel, cleaned seperately with saturated sodium bicarbonate solution, THF and distilled water, dried at 80 °C for 6 h, and then perfomed SEM, XRD, XPS, CF monofilament tensile test and photographing analysis. Meanwhile, the corresponding depolymerization solution and the CF-cloth cleaning liquid were combined together, neutralized with saturated sodium bicarbonate solution, and then filtered, cleaned and dried to gather the precipitate. More experimental details are listed in Supplementary Methods.

**Data availability.** The authors declare that the data supporting the findings of this study are available from the corresponding authors upon request.

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

## Acknowledgements

We thank the support of the National Natural Science Foundation of China (Grants: 51273221 and U1201243), the opening project of Key Laboratory for Polymeric Composite and Functional Materials of Ministry of Education and Science and Technology Program of Guangzhou, China.

## Author contributions

Y.Y. designed the experiments; Y.Y., S.Y., Y.S., X.Z. and L.J. performed the experiments and data analysis; Y.Y. and S.Y. organized and wrote the manuscript; M.Z., J.Z. and S.L. provided valuable insights for the study and commented on the manuscript.

**Additional information**

**Competing financial interests:** The authors declare no competing financial interests.

