## [Peer Review File · Nature Communications]

Reviewers' Comments:

Reviewer #1 (Remarks to the Author):

The reviewer recommends the publication of this paper.

Page 15, line 10 and page S16, line 16
“depolymerizethe” should be “depolymerize the”.

Table 1a and 1b; The acetal epoxy/62vol% (references 15, 16)
“unidirectional CF” should be “cross-ply CF”.

Reviewer #2 (Remarks to the Author):

In this manuscript, Yuan et al. present a fully recyclable composite system employing the resin developed in the labs of Garcia et al., polyhexahydrotriazine (PHT). They successfully demonstrate that the resin system can be used for the formation of composites with carbon fibers and fabrics and that the composite can be completely reverted in acidic conditions. They perform a full characterization on a new PHT using the monomer 2,2-Bis[4-(4-3aminophenoxy)phenyl]propane (BAPP) that shows excellent thermal stability and high degradation temperatures. Furthermore, they show a suite of materials used for forming composites and show through microscopy and other method that PHTs revert without damaging the surface of the carbon fiber. The study is thoughtfully done, thorough and represents advances in the field towards composites and worthy of publication in Nature Communications. I have very minor suggestions for revision: 1) some of the grammar needs to be addressed or edited by a native English speaker for fluidity and accuracy and 2) it would be preferred to specify the PHT resin is the BAPP-PHT in Table 1.

Reviewer #3 (Remarks to the Author):

This paper studied multiply fully recyclable carbon fiber reinforced heat resistant covalent thermosetting advanced composites. This reviewer would think that this work seems to have some uniqueness and medium-level innovation for practical applications of recyclable carbon fiber nondestructively, and thus suggest to accept conditionally with the some corrections as

follows:

- 1) This work used a specific poly(hexahydrotriazine) (PHT) as matrix unlike conventional thermoset epoxy matrix. Defence it to be able to apply epoxy type composites for high temperature application.
- 2) Identify the chemical stability of the carbon fiber using THF of high polar solvent or in THF/HCl solvent. Although some damage of fiber surface may occur, explain your defence how composite properties can not be changed under many recycle number.
- 3) Add the change in the carbon fiber tensile properties (not the property of composites) with many recycle number. Single fiber tensile strength before and after recycled number should be added to identify the thermal and chemical stability of carbon fiber.
- 4) In Figure S9, there are the excess resin adsorbed in carbon fiber surface. Identify how such adsorbed excess resin can change with recycled treatment time and number.
- 5) Identify the chemical reaction at the interface between carbon fiber and PHT matrix during recycling process.
- 6) In figure 5, SD of modulus became rather larger with recycled number. This does not match that mechanical properties of composite does not change with recycled number. Defence about it.

Response to the Reviewers' comments on NCOMMS-16-19631

Response to the comments of Reviewer #1:

The reviewer recommends the publication of this paper. Page 15, line 10 and page S16, line 16 "depolymerizethe" should be "depolymerize the".

Many thanks for the Reviewer's recommendation on publication. The mentioned errors have been corrected in our revised manuscript, respectively.

Table 1a and 1b; The acetal epoxy/62vol% (references 15, 16) "unidirectional CF" should be "cross-ply CF".

Thanks for reminding. These errors have been corrected in Table 1a and 1b in our revised manuscript.

Response to the comments of Reviewer #2:

In this manuscript, Yuan et al. present a fully recyclable composite system employing the resin developed in the labs of Garcia et al., polyhexahydrotriazine (PHT). They successfully demonstrate that the resin system can be used for the formation of composites with carbon fibers and fabrics and that the composite can be completely reverted in acidic conditions. They perform a full characterization on a new PHT using the monomer 2,2-Bis[4-(4-3aminophenoxy)phenyl]propane (BAPP) that shows excellent thermal stability and high degradation temperatures. Furthermore, they show a suite of materials used for forming composites and show through microscopy and other method that PHTs revert without damaging the surface of the carbon fiber. The study is thoughtfully done, thorough and represents advances in the field towards composites and worthy of publication in Nature Communications. I have very minor suggestions for revision: 1) some of the grammar needs to be addressed or edited by a native English speaker for fluidity and accuracy and 2) it would be preferred to specify the PHT resin is the BAPP-PHT in Table 1.

Thank the Reviewer for the positive comments and recommendation on publication very much. As for the language problem, unfortunately, a native English speaker is not available for us, but we invited a proficient English editor to polish the English and correct the grammatical errors, and the fluidity and accuracy should be greatly improved in our revised manuscript. Besides, in order to avoid the confusions between different kinds of PHT resins, we accepted the reviewer's suggestion and specified our PHT resin from BAPP as "the BAPP-PHT" and the PHT resin from ODA by Garcia et al., as "the ODA-PHT" in Table 1 as well as the whole manuscript and supplementary file.

Response to the comments of Reviewer #3:

This paper studied multiply fully recyclable carbon fiber reinforced heat resistant covalent thermosetting advanced composites. This reviewer would think that this work seems to have some uniqueness and medium-level innovation for practical applications of recyclable carbon fiber nondestructively, and thus suggest to accept conditionally with the some corrections as follows: 1) This work used a specific poly(hexahydrotriazine) (PHT) as matrix unlike conventional thermoset epoxy matrix. Defence it to be able to apply epoxy type composites for high temperature application.

We highly appreciate the Reviewer's objective and constructive comments.

As is known to all, epoxy type composites for high temperature application are characterized by excellent comprehensive performances, e.g. high thermal stability, great mechanical property and strong chemical resistance, etc. To verify the applicability of our PHT resin in these highly-demanding fields, we performed thorough characterization on our PHT resin and the related composites with standard or reliable methods and instruments, including the tests of T_g , T_d , tensile strength, Young's modulus, elongation at break, flexural strength and modulus, compression strength, short-beam strength, fracture toughness and chemical resistance. Then we compared the obtained results with those of some representative commercial epoxy matrices. As shown in Table 1b, all characterization data of the BAPP-PHT are comparable to those of the traditional high-performance epoxy resins, so do the CF/PHT composites, indicating its commensurately great comprehensive properties as

epoxy matrix. Moreover, the BAPP-PHT and CF/PHT composite demonstrate thermal stability superior to that of some epoxy resins according to their higher T_g and T_d values (up to 200.1 and 368.5 °C, respectively). Therefore, we plenary believe this PHT resin merit great potential and feasibility for high temperature applications, and the multiple intact CF recyclability of its composite further enhances its application potential and value.

Identify the chemical stability of the carbon fiber using THF of high polar solvent or in THF/HCl solvent. Although some damage of fiber surface may occur, explain your defence how composite properties cannot be changed under many recycle number.

It is the extra-special properties of carbon fiber (CF), such as high stiffness, high tensile strength, low weight, high chemical resistance, high temperature tolerance, low thermal expansion, etc. that make it very popular in many fields. CF is mainly composed by carbon atoms bonded together in dense crystal, and therefore exhibit strong corrosion resistance to various solvents except for some strong oxidants such as H_2O_2 and concentrated sulfuric acid. To confirm its chemical stability to polar solvents, we had soaked it in some high polar solvents such as THF and NMP, and non-oxidizing solutions such as dilute H_2SO_4 /THF and HCl/THF solutions for more than two weeks, then took out and performed characterization test on its physical and chemical structures by various techniques including SEM, XRD, XPS, TGA, FTIR and monofilament tensile test. And no detectable changes were found for any of the treated CFs, strongly identifying its high corrosion resistance to polar solvents.

To investigate whether the CF-PHT composite properties are changed after many recycles, the second and third recycling of the composites were performed sequentially, and then the structures and properties of the twice- and thrice- recycled CFs and the regenerated composites were characterized respectively. As shown in Fig. 5 and supplementary Figs S20-S23, neither the second nor the third recycling circulations impose any detectable influences on the architecture, length, and performance of the CF cloths and the composite mechanical properties. Moreover, the XPS results reflect that no chemical interaction occurs between the PHT matrix resin and the CF cloths (supplementary Fig. S21 and S22). Based on these, we firmly believe that the composite properties will not be changed even under many recycles.

Add the change in the carbon fiber tensile properties (not the property of composites) with many recycle number. Single fiber tensile strength before and after recycled number should be added to identify the thermal and chemical stability of carbon fiber.

We absolutely agree that the tensile strength result of the retrieved carbon fiber is very important and necessary to evaluate the influence of the recycling process on its thermal and chemical stability, therefore, we measured this parameter before and after each recycling circulation and the relevant data are presented detailedly in Fig. 5b and Fig. S23.

In Figure S9, there are the excess resin adsorbed in carbon fiber surface. Identify how such adsorbed excess resin can change with recycled treatment time and number.

The SEM images in Figure S9 present the internal structure of the original CF/PHT composite, including its cross section, tensile and flexural fracture planes, not the recycled ones. Fig. 3c shows how the weight of the adsorbed resin in composites changes with the recycle times. In fact, during the recycling process, the resin is completely degraded and separated from CFs, which is clearly illustrated in Fig. 4, 5 and S20-S22, and there is no adsorbed resin on the surface of CFs after each recycling circulation.

Identify the chemical reaction at the interface between carbon fiber and PHT matrix during recycling process.

In our study, in order to identify whether chemical reaction has occurred at the interface between CFs and PHT matrix during the recycling process, the retrieved CFs of each recycling cycle were respectively characterized by SEM, XRD, XPS, TGA, FTIR and monofilament tensile test, and the results are shown in Fig. 5, S20-S23. However, all these results reveal that no chemical interaction occurs between the BAPP-PHT matrix resin and the CFs.

In figure 5, SD of modulus became rather larger with recycled number. This does not match that mechanical properties of composite does not change with recycled number. Defence about it.

Indeed, as shown in Fig. 5, only SD of modulus apparently becomes rather larger with recycled number, but not those of flexural property and short-beam strength. We attribute this to the test method and the data processing mode. The diameter of single carbon fiber is very small, about 7 μm , therefore, the process of its tensile test is very special. According to the ASTM C1557-14 standard, single carbon fiber is randomly chosen and separated from CF bundle, then is mounted and bonded on a special mounting paper tab via carefully placing a small amount of suitable epoxy adhesive at the marks on the mounting tab. The system compliance is determined experimentally for a given test machine, gripping system and fiber type in advance. As strain is not measured directly, the fiber Young's modulus is obtained from the process used to calculate the system's compliance. In the absence of a direct measurement of specimen elongation, the actual specimen elongation in the gage length is determined by subtracting the displacement associated with the system compliance from the total cross-head displacement. Overall, tensile strength is calculated from the ratio of the peak force and the cross-sectional area of a plane perpendicular to the fiber axis, at the fracture location or in the vicinity of the fracture location, while Young's modulus is determined from the linear region of the tensile stress versus tensile strain curve.

Because the CF diameter is too small, it breaks easily and tensile strain cannot be directly measured. These factors cause the test process to become more complex and lead to a relatively larger standard deviation. This is consistent with the results of the literature (Yamaguchi, A., Hashimoto, T., Kakichi, Y., Urushisaki, M., Sakaguchi, T., Kawabe, K., Kondo, K., Iyo, H. *J. Polym. Sci., Part A: Polym. Chem.* 53, 1052-1059 (2015)). For our tests, at least twenty specimens are successfully tested for each group. Then, twenty valid data are selected to calculate the average and the standard deviation. In fact, in order to master the test method, more than forty origin CF filaments were successfully tested. For the next calculation, we only select twenty test results whose values are very close. As a result, the standard deviation of virgin CFs is relatively small. But for recycled CFs, about twenty five CF filaments were successfully tested and twenty test results were selected for the next calculation. Therefore, their standard deviations seem relatively large, especially for the fiber Young's modulus. Now, we have carried out the tensile test again and processed the data in the same way. For all CFs, thirty specimens were successfully tested for each group and all results were used to calculate the average and the standard deviation. The relevant data has been given in Fig. 5b in the revised manuscript. This matches

that mechanical properties of CFs and composite does not change with recycled number (Fig. 5b and 5c).

Reviewers' Comments:

Reviewer #3 (Remarks to the Author):

Since most of inquiry was answered properly and revised by author, this reviewer suggest this paper to accept with this state.

Response to the Reviewers' comments on NCOMMS-16-19631B

Response to the comments of Reviewer #3:

Reviewer #3 (Remarks to the Author): Since most of inquiry was answered properly and revised by author, this reviewer suggest this paper to accept with this state.

Many thanks for the Reviewer's comments and recommendation on publication.